# Vitamin D3 and Ischemic Stroke: A Narrative Review

**DOI:** 10.3390/antiox11112120

**Published:** 2022-10-27

**Authors:** Władysław Lasoń, Danuta Jantas, Monika Leśkiewicz, Magdalena Regulska, Agnieszka Basta-Kaim

**Affiliations:** Maj Institute of Pharmacology Polish Academy of Sciences, Department of Experimental Neuroendocrinology, Smetna 12, 31-343 Krakow, Poland

**Keywords:** vitamin D3, brain ischemia, neuroprotection, post-stroke depression, neuroinflammation, vascular system, molecular mechanisms

## Abstract

Ischemic stroke is one of the major causes of death and permanent disability worldwide. The only efficient treatment to date is anticoagulant therapy and thrombectomy, which enable restitution of blood flow to ischemic tissues. Numerous promising neuroprotectants have failed in clinical trials. Given the complex pathomechanism of stroke, a multitarget pharmacotherapy seems a more rational approach in stroke prevention and treatment than drugs acting on single molecular targets. Recently, vitamin D3 has emerged as a potential treatment adjunct for ischemic stroke, as it interferes with the key prosurvival pathways and shows neuroprotective, anti-inflammatory, regenerative and anti-aging properties in both neuronal and vascular tissue. Moreover, the stimulatory effect of vitamin D3 on brain-derived neurotrophic factor (BDNF) signaling and neuroplasticity may play a role not only in the recovery of neurological functions, but also in ameliorating post-stroke depression and anxiety. This narrative review presents advances in research on the biochemical mechanisms of stroke-related brain damage, and the genomic and non-genomic effects of vitamin D3 which may interfere with diverse cell death signaling pathways. Next, we discuss the results of in vitro and in vivo experimental studies on the neuroprotective potential of 1alpha,25-dihydroxyvitamin D3 (calcitriol) in brain ischemia models. Finally, the outcomes of clinical trials on vitamin D3 efficiency in ischemic stroke patients are briefly reviewed. Despite the mixed results of the clinical trials, it appears that vitamin D3 still holds promise in preventing or ameliorating neurological and psychiatric consequences of ischemic stroke and certainly deserves further study.

## 1. Introduction

The central nervous system (CNS) is extremely sensitive to a shortage of oxygen and glucose, and a sudden loss of blood circulation to an area of the brain due to systemic hypo-perfusion, thrombosis, or embolism which can result in ischemic stroke. In 1970, the World Health Organization defined stroke as “rapidly developing clinical signs of focal (or global) disturbance of cerebral function, lasting more than 24 h or leading to death, with no apparent cause other than that of vascular origin” [1]. This classic definition of stroke has been updated by the American Heart Association/American Stroke Association, which pointed out that ischemic stroke specifically refers to CNS infarction, which can be defined as “brain, spinal cord, or retinal cell death attributable to ischemia, based on neuropathological, neuroimaging, and/or clinical evidence of permanent injury” [2]. Stroke is considered to be an acute cerebrovascular disease and includes ischemic stroke (about 85–90% of strokes) and hemorrhagic stroke, while the latter is further subdivided into intracerebral hemorrhage and subarachnoid hemorrhage. The current treatment of acute ischemic strokes is based on reperfusion therapies including intravenous administration of thrombolytic agents and endovascular therapy [3,4]. Ischemic stroke is one of the major causes of death and long-term severe disabilities worldwide. According to the World Health Organization, 15 million people suffer from strokes worldwide each year [3]. Moreover, there are estimates that there will be 23 million first-case strokes and 7–8 million stroke deaths in 2030 [4]. Even worse, clinical observations showed that the SARS-CoV-2 pandemic engulfing the entire world in recent years brought about an increase in stroke incidences among COVID-19 patients [5,6,7]. However, a causative link between COVID-19 and stroke has not been proven, yet, and the long-term consequences for cerebrovascular complications among COVID-19 survivors remain unknown [8,9,10].

Despite enormous efforts and substantial investments of the pharmaceutical industry, no clinically efficient and well-tolerated neuroprotective drug has been marketed yet. Nevertheless, it is still expected that a better understanding of pathological events involved in ischemic brain injury, as well as interconnection and crosstalk between various ischemia-induced cell death programs will be helpful in discovering new therapeutic strategies. Because of a high complexity of molecular processes involved in the pathomechanism of stroke, agents with pleiotropic activities rather than those aimed at a single molecular target may be more promising as candidates for neuroprotective drugs. This view is in agreement with the statement of the Stroke Treatment Academic Industry Roundtable (STAIR) which recommended to target multiple mechanisms simultaneously instead of treating a solo pathway after stroke injury [11]. It is proposed that vitamin D3 is one of such multitarget drugs, as it stimulates the key pro-survival pathways and shows neuroprotective, antioxidant, anti-inflammatory, regenerative and anti-aging properties not only in neuronal but also in glia and vascular tissue [12]. The latter is of importance for revascularization of the stroke-affected brain region and recovering normal functions of the blood–brain barrier (BBB), which is regulated by the interactions between the brain endothelium, astrocytes and neurons [13]. Of note, an intimate relationship between the brain and its vessels in the concept of the neurovascular unit has been strongly accentuated [14]. As vitamin D3 stimulates some pro-survival pathways essential for neuroplasticity, including up-regulation of neurotrophin gene expression, it has been suggested that supplementation with vitamin D3 might be beneficial, not only in restoration of neurological functions after stroke, but also in alleviating the post-stroke psychiatric disorders, e.g., depression and anxiety, and improving the deteriorated cognitive functions [15,16]. An increasing body of evidence suggests that vitamin D3 has a positive impact on the prevention of cardiovascular diseases and rehabilitation outcomes in stroke patients; however, due to various methodological limitations of the studies conducted to date it is difficult to draw final conclusions [17]. In this narrative review, first, we present a current view on the pathomechanism of ischemic stroke, and pivotal biochemical mechanisms of ischemic neuronal injury, and then we outline the genomic and non-genomic effects of vitamin D3 which may interfere with ischemia-triggered death signaling pathways. Next, we discuss results of preclinical studies on the neuroprotective effects of 1α,25-dihydroxyvitamin D3 (1,25(OH)2D3, calcitriol), an active form of vitamin D3 in in vitro and in vivo ischemic stroke models; we also summarize available data on clinical trials on vitamin D3 supplementation in stroke patients. Finally, a potential role of vitamin D3 supplementation in promoting brain repair processes and ameliorating post-stroke depression during the rehabilitation period has been briefly described.

## 2. Pathomechanisms of Ischemic Stroke

### 2.1. Pathomorphological Features of Ischemic Stroke 

The ischemia-induced pathophysiological changes in the brain tissue are time-dependent and can be roughly divided into three phases: acute, subacute, and late phase [18]. The acute phase refers to the first 24 h after stroke and its characteristic feature is necrotic death of neuronal and glia cells in the core of the stroke lesions. Pathomorphological analysis shows that the acute irreversible ischemic neuronal injury is characterized by the presence of eosinophilic cytoplasm in the affected neurons, which lacks identifiable substructure, and has a pyknotic or collapsed nucleus. Around the core zone, a much more extended area of moderately ischemic tissue, called ischemic penumbra, can be detected. In this area, insufficiently supplied with blood by collateral arteries, the metabolic and functional activity of neurons deteriorates, but morphological integrity of the cells is still maintained [19]. In the subacute phase, which lasts several days following stroke, the activation of glia cells and neuroimmunological processes appear to play the main role in the extent of brain injury. In the late phase, a glia scar is formed and some reparative processes connected with proliferation and differentiation of cells along with angiogenesis in the stroke-affected brain tissue take place, but inflammation is still evident [20].

### 2.2. Biochemical Basis of Ischemic Stroke and Neuroprotective Strategies of Its Treatment 

A number of biochemical processes have been implicated in the mechanism of ischemia-induced brain damage, including excitotoxicity, oxidative stress, inflammation, acidotoxicity, and apoptosis: Figure 1 [21]. Biochemical changes in ischemic brain tissue are time-dependent and take the form of complex cascade processes [22]. Early neuronal death in the core of the infarct that occurs within a short time after brain damage is most likely due to necrosis, while delayed neuronal death occurring over days and months shows features of an apoptotic process [23]. In the acute phase of the stroke, deprivation of oxygen and glucose inhibits mitochondrial ATP production resulting in neuronal energy deficit and impairment of Na^+^/K^+^-ATPase pumps. These changes cause a disturbance of ion gradients, gradually decrease neuronal membrane potential, and enhance the release of neurotransmitters, mainly represented by glutamate (Glu). Moreover, by reversing Glu, transport ischemia further increases extracellular concentrations of this excitatory neurotransmitter [24,25]. Depolarization of neuronal membrane leads to the activation of voltage-dependent calcium channels and, by removing the magnesium block, it enables the activation of glutamatergic NMDA receptors. Excitotoxicity caused by excessive Glu release, overactivation of NMDA receptors, and increase in intracellular Ca^2+^ concentration, has long been established as a pivotal mechanism of hypoxia-induced neuronal injury [26]. Indeed, the enhanced Ca^2+^ influx into cells is considered to be the main factor activating a cascade of enzymatic processes such as those catalyzed by proteases, lipases and DNases which ultimately destroy intracellular proteins, lipids and nucleic acids, leading to neuronal death. Although NMDA receptors play a key role, ischemia-induced neuronal damage, AMPA receptors and metabotropic mGluR1 receptors are also engaged in this process [27]. However, clinical trials with classical NMDA receptor antagonists conducted to date have been terminated because of undesired side effects and insignificant neuroprotective action of NMDA receptor antagonists, as well as in many cases with a limited therapeutic window [28]. More recently, it has been recognized that synaptic NMDA receptors fulfill physiological functions in synaptic plasticity and cognitive functions, whereas extrasynaptic receptors mediate excitotoxicity [29,30]. Recently, it has been found that extrasynaptic NMDAR subunits GluN2A and GluN2B form a complex with transient receptor potential cation channel subfamily M member 4 (TRPM4) and that this complex is accountable for ischemia-related excitotoxicity, since disruption of the complex provides neuroprotection without disturbing physiological NMDAR-induced calcium signals [31]. These findings, which may open new avenues in the pharmacology of NMDA receptors, await further studies. The oversupply of intracellular calcium ions also induces mitochondrial membrane depolarization, resulting in the release of free radicals which contribute to the complex mechanism of ischemic neuronal damage [32]. It is commonly agreed that in the treatment of acute ischemic stroke, blood supply restoration is critical for salvaging the penumbra-marked brain tissue. However, with restitution of blood flow to ischemic tissues, there is a paradoxical increase in the production of superoxide, nitric oxide and peroxynitrate radicals, which, via activation of matrix metalloproteases, have a damaging effect on BBB integrity, enabling infiltration of neutrophils and leukocytes into the brain parenchyma and promoting neuroinflammation [33]. Overall, an ischemia/reperfusion (I/R) induces complex pathological processes involving intracellular and extracellular pathways that result in metabolic, thrombotic, and inflammatory changes in the affected tissues [34]. While prolonged I/R injury leads to further neuronal cell death which could be executed by apoptotic or non-apoptotic (necroptosis, ferroptosis, parthanatos, and pyroptosis) cell death programs, the moderate injury may favor autophagy and activate recovery systems for survival [35,36,37].

The subacute phase, which occurs within few days after stroke, is mainly characterized by the development of inflammation and activation of apoptosis-inducing signaling pathways. The pathological processes initiated during the acute phase led to the release of some endogenous molecules DAMP (damage-associated molecular patterns) from injured cells, which activate Toll-like receptors, stimulating the release of proinflammatory cytokines and chemokines [38]. Of them, Interleukin-1 beta (IL-1β) and tumor necrosis factor alpha (TNF-α) appear to play a particularly important role in stroke pathology, since they have a damaging effect on neuronal and glia cells and enhance the release of proinflammatory prostaglandins. DAMP stimulate PRR receptors (pattern recognition receptors) resulting in the formation of proinflammatory protein complexes (inflammasomes) [39]. During the subacute phase of stroke, the programmed cell death, apoptosis, prevails in the metabolically impaired penumbra region. Apoptosis, contrary to necrosis, is a slow process, requiring energy supply, switching on the gene transcription, and protein synthesis. Delayed neuronal death in focal cerebral ischemia is characterized by cell shrinkage, chromatin condensation, upregulation of proapoptotic Bcl-2 family members (Bax, Bad, Bak) and activation of caspases [40]. Although many preclinical studies showed neuroprotective efficacy of various anti-apoptotic strategies in brain ischemia models, they have not been specifically tested in the clinic due to insufficient knowledge on the role of apoptosis in stroke and its interplay with other pathological events occurring during ischemic stroke [23,41,42]. The occurrence of delayed cell death during brain ischemia creates favorable conditions for a therapeutic window, giving a chance for timely pharmacological intervention in order to salvage neurons in the area of moderate hypoxia [43,44].

During the long-lasting late phase of ischemic stroke, neuroinflammatory and neuroplastic processes, including neurogenesis, angiogenesis and remyelination are highly active [45]. In this phase, microglia of M2 phenotype are engaged in repair of the brain region damaged by ischemia via production of anti-inflammatory cytokines (IL-4, IL-10), enhancement of phagocytosis of cellular debris and promoting neuroplasticity. The involvement of epigenetic processes, including DNA methylation, post-translational modifications of histone proteins and microRNAs in post-ischemic brain repair and neuroplasticity has also been postulated [46,47].

## 3. The Basics of Vitamin D 

### 3.1. Sources, Biosynthesis and Metabolism

Vitamin D is a fat-soluble vitamin with a pivotal function in the maintenance of calcium and phosphate homeostasis in vertebrates [48,49]. Based on chemical structure, vitamin D belongs to secosteroids and currently its five forms, called D1, D2, D3, D4 and D5, have been identified, which differ in their ultrastructural conformation and origin. The main source of vitamin D in humans (~80–90%) originates from its endogenous non-enzymatic production in the skin epidermis where under sun exposure (UV-B rays, wavelength 290–320 nm) 7-dehydrocholesterol is transformed into an unstable intermediate pre-vitamin D3 which further, in a thermo-sensitive process, isomerizes to vitamin D3 (cholecalciferol) [49,50,51]. About 10–20% of vitamin D could come from the diet (second source) after consumption of plant-derived food enriched in vitamin D2 (ergocalciferol) or animal-based food (e.g., fish oils, fatty fish, beef, liver, eggs or milk) which contain mainly cholecalciferol. Since nowadays a lot of people work indoors and often use sunscreen during sun exposure, their endogenous vitamin D3 production (especially in north latitudes) could be insufficient to maintain health. Thus, various vitamin D supplements (third source) are commercially available, among which the cholecalciferol form predominates as the compound thought to be of a better bioavailability and biological activity [48,49,52,53]. Vitamins D3 and D2 are inactive substances and to evoke their biological effects they need to be activated in a two-step hydroxylation process. The first step takes place in the liver where 25-hydroxylation (mainly via CYP2R1) of cholecalciferol or ergocalciferol leads to the formation of circulating metabolites: 25-hydroxyvitamin D (25(OH)D3 or 25(OH)D2, calcifediol, a clinically used biomarker of serum vitamin D level. In the second step, these metabolites are 1 α-hydroxylated (via CYP27B1) in the kidneys to active forms of vitamin D, 1,25(OH)2D3 (calcitriol) or 1,25(OH)2D2 which possess broad spectrum biological activities [49,50,54]. Since 1,25(OH)2D3 functions as a steroid hormone with a key role in maintaining calcium bone homeostasis, its endogenous production is tightly regulated in the ways typical of the endocrine system, i.e., by feedback inhibition (negative regulation), parathyroid hormone (PTH, stimulant of the renal production of 1,25(OH)2D3), fetal growth factor 23 (FGF-23; inhibits calcitriol production) and by serum concentrations of calcium and phosphate [49,51]. The excessive amount of active (1,25(OH)2D) and storage (25(OH)D) forms of vitamin D are metabolized by 24-hydroxylation (via CYP24A1 to 1,24,25(OH)3D, calcitroic acid or 24,25(OH)2D) to prevent potential toxicity of vitamin D (especially supplemented vitamin D), but some of these metabolites could still possess some biological activities (e.g., 1,24,25(OH)3D in the regulation of bone health). The alternative pathway of vitamin D metabolism involves 20-hydroxylation (via CYP11A1) forming the product 20(OH)D which, together with its metabolite 20,23(OH)2D, are devoid of calcemic activity but possess other biological properties, e.g., anticancer activity [55,56,57]. The amount of free vitamin D metabolites available for biological activity (approx. 0.4% of total 1,25(OH)2D and 0.03% of total 25(OH)D) is also regulated in plasma at the level of their association with DBPs (vitamin D binding proteins). These proteins are characterized by high polymorphism in humans and bind about 58% of the circulating vitamin D metabolites, thus, they are claimed to be responsible for interpersonal differences in vitamin D bioavailability [58]. Other factors which could affect vitamin D metabolism and functions include physical activity, lifestyle, certain medications, environmental pollutants or epigenetics [59].

### 3.2. Genomic and Non-Genomic Mechanisms of Vitamin D Action

Biological action of calcitriol could be executed by genomic and non-genomic mechanisms. The former mechanism is structurally and mechanistically well understood as being the main player in the maintenance of vitamin D homeostasis [12,51,60]. The hydrophobic substance 1,25(OH)2D3 passively penetrates the cell membrane or could be bound to DBPs to be actively transported into cells via endocytosis [61]. When reaching cytosol or nucleus, calcitriol binds to the high affinity vitamin D receptor (VDR). VDR has a molecular weight of 55 kDa and, together with receptors for glucocorticosteroids, sex steroids, thyroxine, retinoids, fatty acids and eicosanoids, belongs to the steroid hormone receptor superfamily. The VDR contains two overlapping ligand binding sites, a genomic pocket (VDR-GP) and an alternative pocket (VDR-AP), that respectively bind a bowl-like ligand configuration (gene transcription) or a planar-like ligand shape (rapid responses) [62]. After vitamin D attaches to the ligand-binding domain, the VDR receptor undergoes heterodimerization, with the retinoic acid X receptor (RXR) initiating a change in its spatial conformation. It has been found that the biologically active form of vitamin D3 promotes the generation of non-genomic responses in the 6-s-cis configuration, while secosteroid binding in the 6-s-trans form may be responsible for genomic responses [63]. After vitamin D binding and translocation to the nucleus, VDR-RXR heterodimer interacts with specific DNA-binding domains, VDREs (vitamin D responsive elements). VDREs are located in the promoter region of vitamin D target genes and are composed of a highly conserved N-terminal DNA-binding domain and alpha-helical C-terminal ligand-binding domain [64]. The presence of several VDREs in the gene promoter suggests that they may act synergistically [65]. Such complexes modulate the activity of RNA polymerase II in the regulatory regions of target genes in various cell types. To promote gene expression, first the co-repressors (e.g., NcoR2/SMART) should be released from VDR/RXR/VDRE complex and then the relevant co-activators (e.g., SRC1) are recruited [12]. The inhibition of gene expression by vitamin D3 could be achieved by direct repression of the transcription of a target gene or indirectly by VDR-mediated enhancement of the transcription of negative regulators of target genes [66]. For genomic mechanisms, attaching the nuclear co-regulator proteins activates or inhibits gene transcription. It should be noted that VDR-binding sites are highly dynamic and could be affected by various factors, such as cell differentiation or maturation state, aging or disease activation and all of these could be reflected by gene expression. The VDR-genomic mechanisms regulate crucial enzymes involved in the synthesis and metabolism of vitamin D3, genes involved in the maintenance of bone calcium homeostasis (for example, genes encoding osteocalcin, osteopontin, 24-hydroxylase enzyme (Cyp24a), 1-hydroxylase enzyme (Cyp27b) or calbindin), as well as genes regulating cell proliferation, differentiation and apoptosis. Such genes include, for example, the genes for p21 protein (cell cycle inhibitor), Bcl-2 protein (regulator of apoptosis), p53 protein (suppressor of oncogenes that control cell growth) and the gene for amphiregulin (epithelial growth factor that stimulates the growth of head, neck and breast cancer cells). Moreover, calcitriol not only has a modulatory effect on growth factor/cytokine synthesis but also regulates growth factor signaling [67]. Additionally, the vitamin D system could be also involved in the regulation of various epigenetic events (e.g., posttranscriptional modifications of histone H3 and H4) and by this way could also affect the transcription of various genes. Reciprocally, various epigenetic modifications influence transcription of the VDR gene and in this way could influence the efficiency and interpersonal differences in vitamin D action [59,68]. Interestingly, in the situation of a low level of 1,25(OH)2D3, the VDR can still operate by binding other molecules, including curcumin, polyunsaturated fatty acids and anthocyanidins, which are thought to be low-affinity nutritional ligands for VDR. Other factors such as resveratrol and sirtuin 1 could potentiate nuclear VDR signaling [51,59].

Apart from the genomic mechanisms, the vitamin D via VDR could mediate its faster biological action when it is distributed outside of the nucleus; however, this area is still not well recognized. Similar to other steroid hormones, this non-genomic action could be connected with VDR localization within the membrane and its interaction, for example, with caveolin 1 (CAV1) and SRC (SRC proto-oncogene) in caveolae to down-regulate WNT, sonic hedgehog (Shh) and NOTCH signaling. The rapid VDR-mediated action may also be a result of its direct interaction with other membrane receptors (eg., calcium channels, mitochondrial permeability transition pore) or intracellular pathways [69]. Recently, a novel mitochondrial localization of VDR has been described as the hub linking the control of cell metabolism [59]. Another non-genomic player for vitamin D3, the enzyme PDIA3 (protein disulphide isomerase family A member 3, also known as ERp57 or 1,25D3-MARRS), is found at various subcellular locations (plasma membrane, endoplasmic reticulum, mitochondria) and up until now is the best described membrane-associated protein that binds vitamin D [12,51,61]. For its action, PDIA3 requires also the interaction with CAV1 and is essential for the activation of protein kinases, such as CaMKII (calcium/calmodulin-dependent protein kinase II), PKC (protein kinase C) or phospolipases (PLA2, PLC) facilitating extracellular Ca^2+^ influx through L-type Ca^2+^ channels (L-VGCC). Other hypothetical non-genomic targets for the vitamin D fast action associated with PDIA3 could include: PKA (protein kinase A), PI3-K, MAPKs (mitogen-activated protein kinases) or Wnt5a (Wnt family member 5A) signaling pathways [12,61,69]. It is also suggested that PDIA3 may not directly bind vitamin D but may serve as a molecular chaperone for VDR or DBPs or other proteins. It should be noted that calcitriol or active vitamin D metabolites (e.g., 20(OH)D3 and 20,23(OH)2D3), when interacting with membrane VDR or PDIA3, could interact with some transcription factors (e.g., STAT3, NF-κB, Nrf2) and in this way they may influence in the long term the expression of various genes. It is not excluded that these active metabolites of vitamin D could also affect other transcription factors, such as RORα (retinoid-related orphan nuclear receptor alpha) and RORγ (retinoid-related orphan nuclear receptor gamma) or AhR (aryl hydrocarbon receptor) which will broaden the complexity of vitamin D action. Although the non-genomic effects of calcitriol and its metabolites, found mostly in in vitro studies, are still not well understood and seem to be dependent on the development stage or are tissue-specific, it is believed that they take place in vivo mainly to fine-tune the VDR-driven genomic response [61]. Further in vivo studies should confirm physiological and clinical significance of the postulated membrane receptors for vitamin D and their non-genomic actions.

### 3.3. Vitamin D Analogues

Over three thousand analogs or mimics of vitamin D were synthetized to overcome calcemic side effects (hypercalcemia and hypercalciuria) of supraphysiological concentrations of vitamin D3 which are needed to evoke pro-differentiating, anti-proliferative or anti-inflammatory effects [70,71,72]. The main modifications of 1,25(OH)2D3 structure are applied to its side-chain, A-ring (often together with side-chain changes), triene system, and C-ring and as a result, the increased VDR binding affinity and higher metabolic stability of the molecules were obtained [72]. However, only a few of the proposed and preclinically tested vitamin D analogs are clinically used to date for the treatment of secondary hyperparathyroidism (alfacalcidol, paricalcitol, doxercalciferol, falecalcitriol, maxacalcitol, oxacalcitriol), psoriasis (tacalcitol, calcipotriol, maxacalcitol) or osteoporosis (alfacalcidol, eldecalcitol) [71,72]. Regarding a potential usage of vitamin D analogs in cancer treatment, despite many promising in vitro results, some agents whichreached clinical trials in acute myeloid leukemia or pancreatic cancer (inecalcitol and seocalcitol, respectively), failed in phase II [72]. This dampened the interest of the pharma industry in further development of vitamin D compounds; however, some research is still in progress in academia in order to fully understand the actions of VDR agonists and antagonists, which hopefully could be further developed for treatment of various human diseases [70]. Apart from potential anticancer properties, some low calcemic vitamin D3 analogues showed protective effects against apoptotic- and oxidative stress-induced cell damage in neuronal cultures [73]. Interestingly, in a model of hydrogen peroxide-induced SH-SY5Y injury, a differential involvement of MAPK/ERK1/2 and PI3-K/Akt signaling in neuroprotective effects of 1,25(OH)2D3 and its low-calcemic analogue—PRI-2191 was found [74]. In differentiated SH-SY5Y cells, both 1,25(OH)2D3 and its structural analogue ZK191784 prevented amyloid-β peptide 1-42-induced toxicity via a sphingosine-1-phosphate/ceramide/p38MAPK/ATF4 signaling pathway [75]. In another study, vitamin D analogues, maxacalcitol, calcipotriol, alfacalcidol, paricalcitol, and doxercalciferol decreased amyloid-β formation and increased amyloid-β degradation. Calcipotriol was also shown to suppress calcium-dependent aggregation of α-synuclein (the key aggregating protein in Parkinson’s disease) by stimulating calbindin-D28k expression in SH-SY5Y neuroblastoma cells [76].

## 4. The Effects of Vitamin D3 in the CNS

Neurosteroids is the group 1,25(OH)2D3 belongs to, since it could be locally synthetized due to the presence of the key rate-limiting enzyme involved in the production of active form of vitamin D3, i.e., 1α-hydroxylase (CYP27B1) in various regions of the rodent and human brain [60,77]. In the brain, vitamin D3 could modulate multiple brain functions by itself or by cross-talking with other steroids signaling molecules, such as estrogens, progesterone or glucocorticoids [53]. In fact, a vast body of knowledge about the importance of sufficient amounts of vitamin D for proper brain development and its well-being during adulthood and during aging has been gathered in the last two decades, coming from experimental studies with vitamin D deficient animals as well as from clinical observations [60,78,79]. Although there are still many gaps in this area regarding, for example, the timing and duration of the critical window throughout life when low vitamin D may have detrimental impact on the brain, a causative link between low neonatal 25-OHD concentrations and an increased risk of schizophrenia [78,79] and hypoxic–ischemic encephalopathy (HIE) [80] has been evidenced.

Regarding the adult brain, it was postulated in the “two-hit hypothesis” that low vitamin D status may rather exacerbate brain lesions-induced by other detrimental events than being harmful by itself [78]. Various experimental studies showed beneficial effects of vitamin D supplementation in various in vitro and in vivo models of neurological or neuropsychiatric diseases at the level of modulation of neurotransmission, neuroprotection, neuroinflammation or neuroregeneration, and the latter was achieved by up-regulation of a wide variety of neurotrophins mainly in astrocytes (nerve growth factor (NGF), neurotrophin 3 (NT-3) or glial-derived neurotrophic factor (GDNF)) [12,60,79,81]. Since VDRs are abundantly expressed in neuronal and glial cells in various brain regions (prefrontal cortex, hippocampus, cingulate gyrus, thalamus, hypothalamus, and substantia nigra), the potential role of vitamin D in the treatment of the central nervous system (CNS) disorders under consideration, e.g., multiple sclerosis, dementia, Parkinson’s disease, depression, schizophrenia, and autism, has been suggested [78,79]. Of note, the distribution of nVDR receptors in the human and rodent brain is very similar [77]. Neurons and glial cells have been shown to contain 1α-hydroxylase, the enzyme that limits synthesis of the active form of vitamin D3, indicating that vitamin D can be synthesized and metabolized locally in the CNS. The highest concentrations of both 1α-hydroxylase and nVDRs are found in the hypothalamus and substantia nigra [77,78].

Alternatively, 1,25(OH)2D3 can also induce rapid non-genomic actions in the CNS via PDIA3 since this transcript is abundantly expressed in neurons, astrocytes, oligodendrocytes, microglia and endothelial cells [82]. However, the precise molecular mechanisms by which vitamin D affects the brain are still unclear and could engage both genomic and non-genomic mechanisms. Some researchers even suggest a predominant role of non-genomic effects of vitamin D in the brain, which may explain its rapid effect on calcium brain homeostasis, neurotransmission, oxidative status or intracellular signaling [12,82]. The non-genomic actions of vitamin D involve rapid response processes that are not dependent on transcriptional gene regulation. It is observed that 1,25(OH)2D3, by affecting the synthesis of neurotransmitters, growth factors and cytokines, modulates many functions of the CNS both during development and in adults [60,83,84]. In vitro and in vivo studies indicate that the synthesis of nerve growth factor (NGF), glial-derived neurotrophic factor (GDNF) and neurotrophin 3 (NT-3) was stimulated by vitamin D3. It also regulates gene expression of the low-affinity NGF receptor, p75NRT [85]. Moreover, 1,25(OH)2D3 was shown to be neuroprotective in an in vitro model of Alzheimer’s Disease through the restoration of Aβ-induced decrease in GDNF level and activation of the phosphatidylinositol 3 kinase (PI3K)/protein kinase B (Akt)/glycogen synthase kinase-3β (GSK-3β) pathway [86]. Both epidemiological data and results from animal experiments suggest that vitamin D3 deficiency may be a significant factor in increasing the risk of multiple sclerosis, diabetes, schizophrenia, and certain cancers, as well as SARS-CoV-2 virus infections [87,88]. Vitamin D deficiency also affects the expression of genes encoding mitochondrial, cytoskeletal and synaptic proteins in the adult rat brain [89] and causes permanent changes in the developing rat brain, disrupting the balance between neural stem cell proliferation and programmed cell death in the offspring [90]. Currently, as opposed to classical aminergic theories, the importance of neuroplastic and neuroinflammatory processes in the pathomechanism of depression, anxiety and cognitive deficits is underlined. Accordingly, regardless of their primary molecular targets, therapeutic efficacy of antidepressant drugs seems to depend on their ability to reverse brain-derived neurotrophic factor (BDNF) deficit and restore normally functioning neuronal networks in the brain structures relevant to these disorders [91,92]. Vitamin D3 shows antidepressant properties in experimental animal models and its mechanism of action is likely to involve the enhanced neurotrophin synthesis, neuromodulatory activity, as well as antioxidant and anti-inflammatory effects [93]. The hypothesis on the therapeutic potential of vitamin D3 in post-stroke depression is supported by experimental data which showed that vitamin D3 improved motor function and attenuated depression-like behaviors in a post-stroke depression model in mice by up-regulation of hippocampal VDR and BDNF expression [15]. Vitamin D’s ability to ameliorate neuroinflammatory processes supports the notion that this compound may prevent development of some psychiatric disorders in post-stroke patients [94].

## 5. The Effects of Calcitriol on Ischemia-Related Neuronal Injury—A Preclinical Evidence

### 5.1. In Vitro Experimental Studies

Only few in vitro studies have demonstrated the neuroprotective potential of 1,25(OH)2D3 in models relevant to ischemic injury (Table 1). In cellular settings, it could be mimicked by using oxygen-glucose deprivation (OGD) or hypoxia models or by exposure of neuronal cells to excitotoxic stimulants. In this regard, Brewer et al. (2001) [95] found that pretreatment with 1,25(OH)2D3 at relatively low concentrations (1–100 nM), but not at higher, nonphysiological concentrations (500–1000 nM) induced downregulation of L-VGCC and protected hippocampal neurons against Glu- and NMDA-induced excitotoxic injury. In another study, pretreatment (24 or 8 h) but not co-treatment with calcitriol at low concentrations (10 and 100 nM), protected the dopaminergic and non-dopaminergic neurons in 10 DIV (days in vitro) primary mesencephalic cell cultures against Glu-induced cytotoxicity and this effect was blocked by a protein and RNA synthesis inhibitor. This pointed to the engagement of the genomic mechanism in neuroprotection mediated by vitamin D3, which was in line with the detected nuclear expression of VDR in neuronal and glia cells. In the same study, a similar neuroprotective effect of pre-treatment with 1,25(OH)2D3 was found against the calcium ionophore, A23187-evoked neuronal cell damage [96]. The genomic mechanisms involved in neuroprotection mediated by vitamin D3 (10 and 100 nM) were also evidenced after chronic exposure of 9 DIV rat cortical neurons in the model of Glu-mediated toxicity where an increased expression of VDR mRNA was detected [97]. Another in vitro study showed that calcitriol reduced Glu-induced excitotoxicity; however, it was in a complex manner dependent on the origin of the neuronal cells, their stage of maturation in culture, the presence of glia cells, and the duration of exposure to the excitotoxic insult [98]. Moreover, this neuroprotective effect of 1,25(OH)2D3 was associated with the inhibition of caspase-3, an effector apoptotic protease. The study by Atif et al. (2009) [99] not only confirmed the neuroprotective potency of calcitriol pre-treatment in the model of Glu-induced cell damage in rat cortical neurons but also showed its synergism with progesterone and association with ERK1/2 activation. Later on, the same research confirmed the potentiating effect of vitamin D on progesterone neuroprotective effects in an OGD model in primary cortical neurons [100]. Recently, Loginova et al. (2021) [101] analyzed the effects of vitamin D3 at different concentrations on hypoxia-induced morpho-functional characteristics of neuron-glial networks in vitro. They found that under hypoxia, cholecalciferol at higher concentration (1 µM) evoked cell death in primary neuronal culture, whereas at lower ones (0.01 and 0.1 µM) it had neuroprotective effect. It was suggested that possible molecular mechanisms of neuroprotective action of vitamin D3 could involve an increased expression of the transcription factor HIF-1α (Hypoxia-inducible factor 1-alpha) and maintaining the BDNF/TrkB expression ratio in the neuronal cells [101].

### 5.2. In Vivo Experimental Studies

Neuroprotective efficacy of vitamin D3 was evaluated in various in vivo models including neonatal hypoxia, focal I/R injury and global cerebral ischemia (Table 2). As pointed out by Stessman and Peeples (2018), neonates are at a particularly high risk of vitamin D deficiency, in part due to the high prevalence of maternal vitamin D deficiency during pregnancy. Infants born to vitamin D-deficient mothers are at a high risk of developing neonatal brain injury, and neonates with hypoxic-ischemic encephalopathy (HIE) tend to be vitamin D-deficient [102]. Regarding neonatal hypoxia models in animals, Lowe et al. (2017b) reported that 1,25(OH)2D3 (0.1 μg/kg/day for 2 weeks) in combination with hypothermia and N-acetylcysteine (NAC) supports functional recovery in both sexes of neonatal rats significantly better than hypothermia alone or hypothermia and NAC in the severe HIE model in rats. These rats performed better on tests of strength and use of affected limb, adaptive sensorimotor skills, motor sequence learning, and working memory and fewer rats in this group had severe hemispheric volume loss [103]. In the rat model of perinatal asphyxia (7-day-old rat pups subjected to hypoxia–ischemia), acute application of calcitriol in a single dose of 2 μg/kg, 30 min after termination of the insult, or subchronic, 7-day post-treatment with calcitriol, effectively reduced brain damage [98].

Regarding models of global and focal cerebral ischemia, Ekici et al. (2009) evaluated the protective effects of calcitriol and dehydroascorbic acid (DHA), a BBB transportable form of vitamin C, against I/R injury on a middle cerebral artery occlusion/reperfusion (MCAO/R) model in rats [104]. The results of this study showed that the combination of vitamin D3 and DHA increased glutathione (GSH) level and superoxide dismutase (SOD) activities in the cortex and corpus striatum, reduced oxidative stress and prevented ischemic brain damage. In another study, it was reported that pretreatment with calcitriol for eight days significantly up-regulated GDNF levels in the cortex and reduced ischemia-induced brain injury induced by the middle cerebral arterial (MCA) ligation in rats [105]. Evans et al. (2018), found that 1,25(OH)2D3 supplementation reduced subsequent brain injury and associated inflammation after ischemic stroke in male C57BL6 mice. Supplementation with 1,25(OH)2D3 (100 ng/kg/day i.p. for 5 days prior to stroke) significantly reduced infarct volume and expression of inflammatory mediators. The authors concluded that prior administration of exogenous vitamin D could attenuate infarct development and exerted acute anti-inflammatory actions in the ischemic and reperfused brain [106]. There is evidence that calcitriol can protect also the spinal cord from I/R injury in rabbits. In this model, calcitriol pretreatment (0.5 μg/kg/7 days, i.p.) lowered malondialdehyde (MDA) levels, reduced myeloperoxidase, xanthine oxidase and caspase-3 activities, but increased catalase levels. The animals pretreated with calcitriol showed better histopathological, ultrastructural, and neurological scores [107]. The importance of adequate vitamin D supplementation in ameliorating the brain injury caused by I/R has been underlined by the authors of a study on Mongolian gerbils [108]. They found that 1,25(OH)2D3 pretreatment reduced the oxidative stress parameters, the expression of NADPH oxidase NOX subunits, and matrix metalloproteinase MMP-9 expression in the cortex and hippocampus of Mongolian gerbils subjected to ten minutes of global cerebral ischemia, followed by 24 h of reperfusion. Other investigators studied the effect of calcitriol on global cerebral ischemia (GCI)-induced neurological deficits and neuronal cell apoptosis in rats. They found that post-GCI administration of calcitriol attenuated brain edema and neurological deficit in rats. Of note, calcitriol also attenuated neuronal apoptosis through marked activation of ERK1/2 pathway [109]. Using the same model, it was found that calcitriol significantly ameliorated the spatial learning and memory impairments and improved the morphological defects in the CA1 area of the hippocampus. Moreover, calcitriol reduced GCI-induced cell apoptosis, reversed the up-regulation of proapoptotic proteins and increased the expression of VDR and activated the ERK1/2 signaling pathway. It was suggested that calcitriol exerted a protective effect against GCI-induced cognitive impairments via inhibition of apoptotic cascade by activating the VDR/ERK1/2 signaling pathway [110]. Very recently it has been reported that calcitriol pretreatment ameliorated severity of brain ischemia in rats subjected to transient MCAO via alterations of NMDA receptor and CYP46A1 gene expression [111]. Fu et al. (2013) found in the MCAO rat model that intraperitoneal administration of calcitriol reduced infarct volumes through the NR3A-MEK/ERK1/2-CREB pathway [112]. Losem-Heinrichs et al. (2004, 2005) studied the effects of 1,25(OH)2D3 and 17β-estradiol alone or in combination on heat shock protein-32 and heat shock protein-27 distribution in the brain after focal cortical ischemia using the photothrombosis model [113,114]. Both proteins are expressed in the brain as a response to cerebral oxidative stress; however, HSP-32 is engaged in metabolizing toxic free-heme to carbon monoxide, iron and biliverdin, whereas HSP-27 is an inhibitor of apoptosis. In this experiment, only the combined treatment with 1,25(OH)2D3 and 17β-estradiol showed synergistic protective effects. Other synergistic action of vitamin D3 with another steroid hormone, progesterone, was reported by Atif et al. (2013). They found that 1,25(OH)2D3 potentiated progesterone neuroprotective effects in a transient MCAO model in rats [100]. Another group of researchers studied the effects of 1,25(OH)2D3 on HO-1 expression in a model of photothrombosis-induced focal cortical ischemia [115]. In that study 1,25(OH)2D3 (4 μg/kg body weight) did not prolong ischemia-induced HO-1 expression but reduced reactive gliosis in the lesioned brain region. Using the same model of stroke, Oermann et al. (2006) revealed that the post-ischemic neuronal cyclooxygenase-2 (COX-2) upregulation did not contribute to the neuroprotective mechanism of calcitriol [116]. Balden et al. (2012) observed in female rats that an 8 week vitamin D deficient (VDD) diet, which reduced circulating vitamin D levels to 22% of that observed in rats fed control chow, worsened MCAO-evoked brain injury and dysregulated the post-stroke inflammation. However, acute treatments (4 h after stroke and every 24 h thereafter) had no positive effect on infarct volume or behavioral deficit. Cortical and striatal infarct volumes in animals fed VDD diet were significantly larger, and sensorimotor behavioral testing indicated that VDD animals had more severe post-stroke behavioral impairment than controls [117].

Regarding the effects of calcitriol on the vascular system, Won et al. showed that this compound prevented hypoxia/reoxygenation-induced BBB disruption through the upregulation of the expression of the tight junction proteins (zonula occludin-1, claudin-5 and occludin) inhibition of reactive oxygen species (ROS) production, MMP9 expression and NF-κB activation in bEnd.3 cells [118]. I/R-induced BBB impairment is thought to increase the possibility of hemorrhagic transformation, vasogenic brain edema and neuroinflammation. Sadeghian et al. (2019) observed, in the rat model of stroke, that calcitriol reduced cerebral infarct volume, attenuated brain edema formation and improved BBB function [119]. These beneficial effects of calcitriol were associated with the upregulation of antioxidant enzyme activities, diminished cell apoptosis and increased BDNF immunoreactivity in the brain tissue of rats after I/R. Calcitriol may also exert beneficial effects on angiogenesis in post-ischemic brain tissue. To this end, Bao and Yu (2018) showed that calcitriol increased the score of neurological function, decreased the size of cerebral infarction and enhanced cerebral perfusion in MCAO model of stroke in rats. Moreover, these researchers found that calcitriol via activating the sonic hedgehog (Shh) signaling pathway, up-regulated vascular growth-related factors, elevated micro-vessel density after cerebral infarction and promoted the proliferation of vascular endothelial cells in the ischemic cortex [120]. Although vitamin D penetrates the BBB, pharmacological efficacy of this compound may depend on its pharmaceutical formulation. Kumar et al. (2020) [121] investigated efficiency of vitamin D3 nanoemulsion through radiometry, gamma scintigraphy in a transient MCAO rat model. They demonstrated that vitamin D3-nanoemulsion (a mean size range of 49.29 ± 10.28 nm) had a better permeation, deposition, and efficacy through the intranasal route when compared to intravenous injection of vitamin D3 or vitamin D3 nanoemulsion. When considering new neuroprotective strategies, effects of vitamin D not only on cerebrovascular, but also on the peripheral vascular system, should be taken into account. Hypertension is the leading risk for stroke. Experimental research found that calcitriol improved endothelium-dependent vasodilation in rat hypertensive renal injury by enhancing Klotho expression and suppressing oxidative stress [122]. Khan et al. (2018) used nanomedical systems to investigate the role of 1,25(OH)2D3 in the preservation/restoration of endothelial function in an angiotensin II (Ang II) cellular model of hypertension. In this model, the beneficial effect of vitamin D3 was associated with a favorable rate of cytoprotective nitric oxide (NO) and cytotoxic peroxynitrite release and the decrease in the overexpression of inducible nitric oxide synthase (NOS) and NADPH oxidase [123]. Collectively, there is growing pre-clinical evidence supported by biochemical, behavioral and histological studies that vitamin D3 may have protective effects on structure and functioning of the vascular system and BBB integrity in ischemia-related cerebrovascular pathologies.

**Table 2 antioxidants-11-02120-t002:** Effects of vitamin D in experimental stroke models—in vivo studies.

Model	Animal Treatment	Effects of Vitamin D	Reference
Hypoxia ischemia (HI) rat model	Hypothermia treatment + NAC (50 mg/kg) + 1,25-(OH)2D3 (0.1 μg/kg/)/daily for 2 weeks	↑motor skills ↓anxiety ↑spatial learning	[103]
Rat model of perinatal asphyxia in 7-day-old pups	1,25(OH)2D3 (2 μg/kg, i.p., single dose)/30 min after the insult or for 6 consecutive days	↓brain damage	[98]
MCAO/R rat model of I/R injury	1,25(OH)2D3 (1 μg/kg i.p.)/day/8 days before ischemia. DHA (250 mg/mL, from tail vein/30 min before MCAO/R	↓MDA ↑GSH, SOD activity in cortex and corpus striatum in 1,25(OH)2D3 + DHA group	[104]
MCA ligation model in rat	1,25(OH)2D3, 1 μg/kg/day, i.p., 4 or 8 days	↓the amount of infarction in the cortex, ↑GDNF levels	[105]
MCAO/R model in C57BL6 mice	1,25(OH)2D3 (100 ng/kg, i.p./day/5 day prior to stroke	↓infarct volume ↓pro-inflammatory mediators IL-6, IL-1β, IL-23a, TGF-β and NADPH oxidase-2	[106]
Spinal cord I/R injury in rabbit	1,25(OH)2D3 (0.5 μg/kg, i.p./7 days before I/R)	↓MDA, myeloperoxidase, xanthine oxidase activities, caspase-3 level, ↑catalase level, histopathological, ultrastructural, and neurological scores	[107]
BCAO model in Mongolian gerbils	1,25(OH)2D3 1 μg/kg, i.p./day/7 days prior to ischemia.	↓MMP-9 ↓lipid peroxidation ↓superoxide anion production ↑VDR expression	[108]
GCI model in rat	1,25(OH)2D3 (1 μg/kg, i.p.)/30 min, 12 and 24 h after the GCI insult and PD98059 (5 μg, through the tail vein)/30 min prior to the insult	↓brain edema, ↑neurological function, ↑ERK 1/2 pathway activation, ↓neuronal apoptosis	[109]
GCI model in rat	1,25(OH)2D3 (1 μg/kg, i.p.)/30 min, 12 and 24 h after the GCI insult and PD98059 (5 μg, through the tail vein)/30 min prior to the insult	↑the spatial learning and memory ↑neurological function ↓brain edema, ↓morphological defects in the CA1 area of the hippocampus ↓apoptosis ↑ VDR expression ↑ERK 1/2 pathway activation—PD98059 reversed the anti-apoptotic effect of 1,25(OH)2D3	[110]
MCAO model in rat	1,25(OH)2D3, 7 days prior to stroke induction	↓lesion volume, ischemic neurobehavioral deficits, regulation of the glutamate receptor expression and CYP46A1 genes	[111]
MCAO model in rat	1,25(OH)2D3 i.p., (a single dose of 2 μg/kg, immediately following ischemia) and subchronically (2 μg/kg on 6 consecutive days).	↓infarct volumes 7 days following reperfusion, ↑NR3A and CREB activity in the hippocampal neurons, protection of the brain from I/R injury through the NR3A-MEK/ERK1/2-CREB pathway	[112]
Focal cortical ischemia (photothrombosis model) in rat	Lesioned rats were injected i.p. one hour after injury with either 1 μg 1,25(OH)2D3/kg or 7 μg 17β-estradiol/kg or a combination of both steroids	↓HSP-27 within the infracted cerebral cortex	[114]
tMCAO model in rat	Progesterone (8 mg/kg), 1,25(OH)2D3 (1 μg/kg body weight/day) alone or in a combination, 5 min. i.p. prior to reperfusion followed by daily s.c. injections for 6 days.	↓motor deficits, infarct reduction, ↑BDNF, TrkB and p-ERK1/2 expression, ↓apoptosis (↑Bcl-2, ↓caspase-3) ↓IL-6 and p-NF-κB ↑HO-1	[100]
Focal cortical ischemia (photothrombosis model) in rat	Postlesional treatment with 1,25(OH)2D3 (4 μg/kg i.p.)	↑glial HO-1 ↓GFAP	[115]
Focal cortical ischemia (photothrombosis model) in rat	1,25(OH)2D3 (4 μg/kg i.p.)	no significant differences between 1,25(OH)2D3-treated and solvent-treated lesioned rats in neuronal COX-2 expression	[116]
MCAO model in female rat	Vit. D deficiency (VDD) diet for 8 weeks before MCAO; 10 μg/kg 1,25(OH)2D3, 4 h after MCAO and every 24 h thereafter for 5 days	VDD diet effects: ↑cortical and striatal infarct volumes, ↑severe poststroke behavioral impairment ↓IGF-I in plasma and the ischemic hemisphere, ↓IL-1α, IL-1β, IL-2, IL-4, IFN-γ, and IL-10 expression in ischemic brain tissue, ↑IL-6 Acute 1,25(OH)2D3 treatment did not improve infarct volume or behavioral performance	[117]
Hypoxia/reoxygenation (H/R) model in bEnd.3 cells	1,25(OH)2D3 (5–200 nmol/L)/24 h before H/R, continued throughout the H/R period	↑BBB function, zonula occludin-1, claudin-5, and occludin, ↓NF-κB ↓MMP-9	[118]
MCAO model in rat	1,25(OH)2D3 i.p. one group—12 μg/kg immediately after the ischemia period (60 min) second group—2 μg/kg after MCAO and over the next 5 days	↓brain infarction volume, brain edema formation ↑BBB function ↑antioxidant enzyme activities ↓cell apoptosis ↑BDNF immunoreactivity	[119]
MCAO rat model	Vit. D3, 1000 IU/kg/day through gavage/14 days	↓the size of cerebral infarction, ↑cerebral perfusion in the ischemic area ↑levels of vascular growth-related factor ↑micro-vessel density after cerebral infarction and ↑the proliferation of vascular endothelial cells in the ischemic cortex ↑Shh signaling in the ischemic cortex	[120]
MCAO rat model	Vit. D3 100 ng/kg i.v., vit. D3 nanoemulsion i.v. and intranasal (a mean size range of 49.29 ± 10.28 nm, equivalent to 100 ng/kg vit. D3	↑BBB permeation, deposition, and efficacy of vit. D3-nanoemulsion through the intranasal route in comparison i.v. vit. D3 or vit. D3 nanoemulsion	[121]
MCAO model in C57BL6 mice	1,25(OH)2D3 (100 ng/kg, i.p./day/5 days before MCAO	↓the volume of cerebral infarction ↓IL-6, IL-1β, IL-23a, TGF-β, Gp91phox	[106]
MCAO combined with CUMS in mice	Vit. D3 (6–50 μg/kg), icv/4 weeks	↓motor dysfunction and depression-like behaviors ↑VDR expression and BDNF	[15]
tMCAO model in rat	Calcitriol 1 µg/kg, i.p., 7 consecutive days before experimental induction of stroke	↓infarction volume ↓neurological deficits in brain, ↓MDA and NO levels ↑TAC level, ↑HO-1 and Nrf2 protein and mRNA	[124]

↑: increase; ↓: decrease; 1,25(OH)2D3: calcitriol; BBB: blood brain barrier; BCAO: bilateral common carotid occlusion; BDNF: brain derived neurotrophic factor; bEnd.3: an immortalized mouse brain endothelial cell line; COX-2: cyclooxygenase 2; CUMS: chronic unpredicted mild stress; DHA: dehydroascorbic acid; ERK1/2: extracellular signal-regulated kinase 1 and 2; GCI: global cerebral ischemia; GDNF: glial cell-derived neurotrophic factor; GFAP: glial fibrillary acidic protein; Gp91phox: the NOX family of NADPH oxidases; GSH: glutathione; HO-1: heme oxygenase; i.p.: intraperitoneal; I/R: ischemia/reperfusion; MCA: middle cerebral arterial; MCAO/R: middle cerebral artery occlusion/reperfusion model; MDA: malondialdehyde; MEK: mitogen-activated protein kinase; MMP9: matrix metalloproteinases 9; NAC: *N*-acetylcysteine; NO: nitric oxide; NR3A: NMDA receptor subunit 3A; Nrf2: nuclear factor erythroid 2-related factor 2; p-CREB-Ca^2+^: response element binding protein; PD98059: p-ERK1/2 inhibitor; Shh: sonic hedgehog pathway; SOD: superoxide dismutase; TAC: total antioxidant capacity; TrkB: tyrosine receptor kinase B; VDR: vitamin D receptor.

### 5.3. The Effects of Calcitriol on Inflammatory Response in Ischemic Brain

As reviewed by Yang et al. (2019), a large number of neuroinflammatory factors including cytokines, chemokines, oxidative and nitrosative stress, adhesion molecules, matrix metalloproteinases, and vascular endothelial growth factor contribute to disruption of the BBB integrity and neurological deficits in ischemic stroke. However, in the early phase of ischemic stroke, neuroinflammation associated with the release of proinflammatory cytokines is detrimental, whereas at late stages it promotes angiogenesis and neuroregenerative processes in the post-ischemic brain [125]. Vitamin D3 may have beneficial effect in both early and late phases of stroke because of its ability to decrease proinflammatory status in the brain by diminishing immune cell trafficking through BBB and inhibiting microglial and astrocytic activation [126], and stimulating neuroplasticity. Evans et al. (2018) reported that 1,25(OH)2D3 supplementation (100 ng/kg/day, i.p. for 5 days) reduced the volume of cerebral infarction after ischemia/reperfusion (I/R) in male C57BL6 mice in a MCAO model. Furthermore, 1,25(OH)2D3 reduced inflammatory response as evidenced by a decrease in the expressions of IL-6, IL-1β, IL-23a, TGF-β and NADPH oxidase-2 in the ischemic brain tissues. The authors emphasized the significance of the anti-inflammatory mechanism of vitamin D in alleviating the development of cerebral infarction after MCAO reperfusion [106]. There is also evidence for the role of vitamin D3 in regulating inflammatory activity within vascular walls [127]. The findings supporting the role of vitamin D in preventing or limiting the development of stroke and cerebrovascular malformations emphasizing the involvement of neuroinflammatory processes, has been recently reviewed by Kim et al. (2020). The authors stressed the significance of vitamin D deficiency or its key downstream effects, including defective autophagy and abnormal pro-oxidant and pro-inflammatory responses in these cerebrovascular diseases [128]. In organotypic hippocampal slices exposed to lipopolysaccharide (LPS)—a model of neuroinflammation—calcitriol in both free and nanoparticle forms suppressed the LPS-induced nitric oxide release and prevented neuronal damage [129]. The multidirectional and multitarget mechanism of vitamin D3 actions is evident not only in the neuronal and glia cells, but also in the immune and vascular brain systems. Vitamin D3 was shown to protect both neuronal and endothelial cells against oxidative stress-related damage [130,131]. El-Atifi et al. (2015) reported a significant upregulation of genes coding for Cyp27B1, which catalyzes the last step of 1,25D synthesis, and of Cyp2R1, that catalyzes the first enzymatic reaction of the vitamin D metabolism in human brain pericytes challenged with inflammatory cytokines TNF-α and interferon-γ (INFγ) [132]. These data suggest the existence of an autocrine/paracrine vitamin D signaling system in the neurovascular unit, which may be important for the control of neuroinflammation and brain pathologies. Astrocytes and pericytes play a crucial role in structure and functioning of the BBB, which is defective in stroke patients. Transcriptomic analysis of human brain pericytes in culture exposed to 1,25(OH)2D3 revealed that these cells responded to this treatment via genes which control neuroinflammation. It is suggested that neuroinflammation could trigger the local synthesis of the active form of vitamin D3 by brain pericytes, which in turn respond to the secosteroid by a global anti-inflammatory response. These results point to cerebral pericytes as calcitriol-responsive cells and reinforce the rationale for potential utility of vitamin D3 in the prevention and treatment of stroke and chronic neurodegenerative diseases [133].

## 6. Vitamin D3 Deficiency as a Risk Factor for HIE and Stroke Severity and Outcomes—Clinical Studies

It has been hypothesized that vitamin D metabolism would be dysregulated in neonatal HIE and that this pathological condition alters specific cytokines involved in Th17 activation, which might be mitigated by hypothermia. A study showed an insufficiency of 25(OH)D, which was observed after birth in 70% of 50 infants with moderate to severe HIE and serum 25(OH)D positively correlated with anti-inflammatory cytokine IL-17E in all HIE infants. However, hypothermia did not mitigate vitamin D deficiency in those HIE neonates [80]. The role of vitamin D as an adjuvant therapy for management of HIE was evaluated in a study in 60 neonates with HIE grade II, divided into two groups. Group I received a single daily oral dose of vitamin D3 (1000 IU, stabilized aqueous solution of cholecalciferol) for 2 weeks, in addition to daily subcutaneous (s.c.) human recombinant erythropoietin (2500 IU/kg) for 5 days and i.m. or i.v. magnesium sulfate 250 mg/kg, whereas Group II received erythropoietin and magnesium sulfate, similar to group I, but without vitamin D. The obtained data showed that vitamin D brought improvement for the group I cases, as demonstrated by the reduction of serum S100-B levels after vitamin D therapy [134].

Due to limited mobility, malnutrition, and low sunlight exposure, vitamin D deficiency affects about 71% of ischemic stroke patients [135]. A number of clinical reports suggest that serum vitamin D3 deficiency can be considered a risk factor for stroke severity and short- and long-term outcomes (Table 3). To this end, Turetsky et al. (2015) retrospectively analyzed in 96 stroke patients whether low serum 25(OH)D level, a marker of vitamin D status, is predictive of the ischemic infarct volume and whether it relates to a worse outcome. They found that higher serum 25(OH)D concentrations were associated with smaller infarct volumes and that the association of 25(OH)D with ischemic infarct volume was independent of other known predictors of the infarct extent. Furthermore, the risk for a poor 90-day outcome doubled with each 10 ng/mL decrease in serum 25(OH)D. It was suggested that vitamin D could be a promising marker for cerebral ischemic vulnerability and to identify stroke patients at high risk for poor outcome [136]. Another study which enrolled a total of 818 consecutive patients with acute ischemic stroke revealed that serum 25(OH)D levels in patients with good outcomes were significantly higher than those with poor outcomes. The authors concluded that serum 25(OH)D level was an independent predictor of functional outcome in patients with acute ischemic stroke [137]. Alfieri et al. (2017) investigated whether VDD was associated with acute ischemic stroke, inflammatory markers, and short-term outcome. This study enrolled 168 acute ischemic stroke patients and 118 controls. Results of this study showed that stroke patients had a higher frequency of VDD than controls. Furthermore, the increased high-sensitivity C-reactive protein level in patients with VDD suggests an important role of vitamin D in the inflammatory response to brain ischemia [138]. A 1-year follow-up study in 387 patients with first-ever ischemic stroke reported a negative correlation between serum 25(OH)D level and infarct volume measured by diffusion-weighted imaging. These findings and further analysis of the mortality distribution led to the conclusion that the serum levels of 25(OH)D at admission can be a prognostic marker of cardiovascular disease and all-cause mortality in Chinese patients with ischemic stroke [139]. Reduced serum 25(OH)D predicts the risk of stroke recurrence in ischemic stroke patients. To this end, in a study on 349 patients with first-ever ischemic stroke, serum concentrations of 25(OH)D were negatively associated with the stroke recurrence. The authors hypothesized that the increased risk of stroke recurrence might be linked to the association of vitamin D deficiency with the presence of hypertension, diabetes mellitus, atherosclerosis as well as hyperparathyroidism-mediated cardiovascular events [140]. More recently, in a cross-sectional study which enrolled 982 ischemic stroke patients, it was found that deficiency of vitamin D was associated with severity of stroke as assessed by the National Institute of Health Stroke Scale in Chinese patients, and that the risk factors of VDD in stroke patients included the female gender and higher blood fibrinogen level [141]. In another randomized controlled single-blind study including 90 subjects, a negative correlation was found between the serum 25(OH)D levels and the severity of ischemic stroke as assessed by the National Institutes of Health Stroke Scale; however, it was not associated with changes in leukocyte DBP and VDR expression [142]. Wajda et al. (2019) investigated the association between deficiency of 25(OH)D and increased risk of mortality in 240 patients with ischemic stroke [143]. They found that severe vitamin D3 deficiency was a strong negative predictor for survival after ischemic stroke, independent of age and functional status. However, they also stated that it was still uncertain whether vitamin D3 supplementation might improve the survival of ischemic stroke patients. In a single center study on a rather small population (N = 235), a significant association between 25(OH)D and infarct volume and stroke severity was reported. As suggested by the authors, mechanisms of the association of low 25(OH)D3 level with stroke severity may include dysregulation of the ischemia-induced inflammation, suppression of neurotrophic factor synthesis e.g., insulin-like growth factor 1(IGF-1), reduction of tissue plasminogen activator secretion, and attenuation of nitric oxide-mediated vasodilatation [144]. In a prospective population-based study including 9680 participants (adjusted for age, sex, study cohort, season of blood sampling, and other cardiovascular risk factors) an association between serum vitamin D level and prevalent stroke was found, but only severe VDD was associated with incident stroke. The authors suggested that lower vitamin D levels did not lead to a higher stroke risk but might be a consequence of stroke due to reduced exposure to sunlight, diet quality, and other factors [145].

Some studies focused on evaluating the efficacy of vitamin D3 supplementation in the rehabilitation period of stroke patients. In this regard, results of the retrospective study conducted in Turkey in 120 stroke patients showed that high levels of vitamin D3 were associated with a greater motor functional gain and lesser cognitive impairment during the rehabilitation program in both ischemic stroke patients and hemorrhagic stroke patients [146]. Sari et al. (2018) [147] evaluated the effect of vitamin D supplementation on rehabilitation outcomes and balance in 132 ischemic stroke hemiplegic patients with vitamin D deficiency. The results of this rather short follow-up (3 months) study demonstrated that vitamin D supplementation increased the activity levels and accelerated balance recovery but had no significant effect on ambulation and motor recovery. Additionally, another short and small-population follow-up study showed that vitamin D supplementation in patients with stroke may increase rehabilitation success in terms of improving lower extremity motor function and ambulation status, but only during the first 3 months post-stroke [148]. Momosaki et al. (2019) [149] studied the effects of vitamin D3 supplementation on outcomes in hospitalized patients undergoing rehabilitation after acute stroke. This seven-month multicenter, randomized, double-blind, parallel-group trial, enrolling 100 patients, revealed that the effects of oral vitamin D3 supplementation (2000 IU/day) did not differ from placebo in rehabilitation outcomes estimated by Barthel Index scores after acute stroke. In order to explore whether there was an association between vitamin D status and the risk of stroke, the study’s authors conducted a systematic review and a meta-analysis by searching three databases: Pubmed, Embase, and the Cochrane Library. Nineteen studies were included and further analysis revealed that vitamin D status was associated with ischemic stroke. The studies included in this meta-analysis with different cutoff values revealed 62% increased risk of ischemic stroke in participants with lower 25(OH)D serum levels [150]. The hypothesis, that vitamin D insufficiency would be associated with a higher risk of poor functional outcomes in nondiabetic stroke patients, was also verified. The results of the study which enrolled 266 nondiabetic subjects with stroke showed that VDD (serum 25(OH)D, 20 ng/mL) was associated with an increased risk of poor functional outcome events in Chinese nondiabetic stroke sufferers [151]. More recently, Yarlagadda et al. (2020) [152] in a systematic review summarized findings from studies relevant to the relationship between vitamin D and stroke. It has been concluded that vitamin D deficiency is a significant risk factor for ischemic stroke, stroke severity and short- and long-term outcomes. An observational Danish study on 11,6655 white individuals, genotyped for genetic variants in DHCR7 and CYP2R1 affecting plasma 25(OH)D, found that low 25(OH)D concentrations were associated with a higher blood pressure and hypertension. However, a causal relationship between low 25(OH)D level and ischemic stroke could not be proven or excluded in this study [153]. In the ‘Reasons for Geographic and Racial Differences in Stroke (REGARDS) Study’, with 610 participants who developed incident stroke, and in 937 stroke-free individuals, it was found that lower 25(OH)D concentrations were an independent risk factor for incident stroke irrespective of black or white race. However, as the authors indicated in their report, standard 25(OH)D assays did not discriminate between relatively inert vitamin D bound to DBP and the biologically active free vitamin D [154].

The neuroprotective mechanisms by which vitamin D operates to mitigate stroke onset and outcomes, include promotion of certain neuroprotective growth factors, reduction of arterial blood pressure through vasodilation, and inhibition of ROS.

The results of vitamin D supplementation in stroke patients are mixed, but merit attention. VDD may have also prognostic value, especially for women and individuals with a DBP single-nucleotide polymorphism (SNP), particularly involving the G allele of rs7041 and A allele of rs4588, which are linked with low 25(OH)D3 levels [155]. It has been suggested that future research should expand the understanding of the neuroprotective mechanisms of vitamin D and should establish how supplementation could be administered effectively in stroke treatment [152]. The randomized, placebo-controlled VITAL trial, comparing the use of vitamin D supplements and marine omega-3 fatty acids with placebo in the elderly, found no decrease in the incidence of cardiovascular events, including stroke, in the intervention group during the mean follow-up of 5.3 years. The authors concluded that any benefit of vitamin D supplementation might have an impact on stroke risk, only in the severely VDD individuals [156]. Recently Zelzer et al. (2021) investigated the association between vitamin D metabolites, cognitive function and brain atrophy in a cohort of well-characterized community-dwelling elderly individuals with normal neurological status and without history of stroke and dementia. The results of this study showed that only a sub-group of individuals with low concentrations of 25(OH)D and 24,25(OH)2D3 showed significantly worse memory function compared to individuals with its normal or high concentrations. It was concluded that VDD individuals appeared to have a modest reduction of memory function without significant MRI-derived indices of neurodegeneration and vascular changes [157]. Post-stroke depression (PSD) and impairment of cognitive functions are frequent and difficult-to-treat brain disorders [158]. Indeed, depression-like symptoms are observed in 12–72% of stroke survivors [159,160]. The low serum levels of vitamin D3 is associated not only with a higher risk of stroke [150] but also with post-stroke depression and anxiety, as the stroke patients who had 25[OH] D deficiency (<20 ng/mL) or insufficiency (20–30 ng/mL) within 24 h of entering a stroke unit were more likely to show PSD at 1 month or 6 months later [161]. Wu et al. (2016) [162] conducted study on 226 first acute ischemic stroke patients and found that low serum levels of vitamin D (≤38.48 nmol/L) were independently associated with the development of post-stroke anxiety. However, in the 12-week, randomized, double-blind, parallel, monocentric clinical trial of 40 patients undergoing intensive neuro-rehabilitation treatment, it was found that the beneficial effect on mood and functional recovery was mainly due to neurorehabilitation rather than vitamin D supplementation. Nevertheless, patients who were treated with 2000 IU/day of oral cholecalciferol showed a more evident improvement [16]. All in all, the majority of clinical reports indicate that vitamin D3 deficiency can be considered a risk factor for stroke severity and its short- and long-term outcomes. On the other hand, the data on efficacy of vitamin D3 supplementation in improvement of stroke outcome, including neurologic and psychiatric conditions in the rehabilitation period, are still inconclusive. Clearly, more clinical well-designed studies are needed to elucidate the efficacy of vitamin D3 in the stroke treatment and post-stroke rehabilitation.

**Table 3 antioxidants-11-02120-t003:** Effects of vitamin D in patients with stroke—supplementation/blood level/clinic symptoms/treatment/healthy control—clinical studies.

Objective	Participants	Vitamin D Administration/ Determination	Effect	Reference
vit. D metabolism in neonatal HIE and involvement cytokines related to Th17 function (*)	50 HIE infants	serum samples from a multicenter randomized controlled trial of hypothermia 33 °C for 48 h after HIE birth vs. normothermia	↓25(OH)D after birth in 70% of infants ↓IL-17E in all HIE neonates	[80]
vit. D as an adjuvant therapy for management of neonatal HIE	60 HIE grade II neonates	vit. D3 (1000 IU, oral)/day/2 weeks and human recombinant erythropoietin (2500 IU/kg/s.c.)/day/5 days and magnesium sulphate 250 mg/kg/i.m. or i.v. half an hour of birth, and subsequently 125 mg/kg/24 and 48 h of life	before therapy: ↓serum 25(OH)D ↑serum S100-B after vit. D: ↓serum S100-B level	[134]
associations between serum 25(OH)D level and the ischemic infarct volume and long-term outcome	96 AIS patients retrospective study	serum 25(OH)D level; calculation the volume of cerebral infarction	low 25(OH)D associated with higher infarct volumes and worse outcome	[136]
associations between serum 25(OH)D level and the functional outcome	818 AIS patients	serum 25(OH)D level	↑serum 25(OH)D in patients with good outcomes	[137]
associations between VDD and inflammatory markers, and short-term outcome	168 AIS patients and 118 controls	serum 25(OH)D, IL-6, TNF-α, hsCRP level	in AIS patients: ↓25(OH)D, ↑frequency of VDD, ↑inflammatory markers (IL-6, hsCRP)	[138]
associations between 25(OH)D serum level and cardiovascular disease (CVD) or all-cause mortality	387 patients with ischemic stroke	serum 25(OH)D	negative correlation between 25(OH)D and infarct volume	[139]
association between 25(OH)D serum level and severity of stroke	986 stroke patients (629 males, 357 females)	serum 25(OH)D, apolipoprotein A-I, apolipoprotein B, ApoA-I/ApoB, cholesterol, fibrinogen, blood glucose, high-density lipoprotein, low-density lipoprotein cholesterol, triglyceride	female gender and higher blood fibrinogen level as the risk factors of VDD and higher severity of stroke	[141]
association between serum 25(OH)D and stroke severity	90 ischemic stroke patients, 39 controls	serum 25(OH)D, DBP and VDR gene expression in leukocytes	negative correlation 25(OH)D levels and the severity of ischemic stroke no changes in DBP and VDR	[142]
associations of 25(OH)D with risk of mortality	240 consecutive patients admitted within the 24 h after the onset of IS	serum 25(OH)D	severe VDD strong negative predictor for survival after IS	[143]
association of serum 25(OH)D with prevalent and incident stroke—population-based study	9680 participants	serum 25(OH)D	severe VDD associated with a higher stroke risk	[145]
association between 25(OH)D and functional outcomes in stroke patients	120 ischemic and hemorrhagic stroke patients participating the neurological rehabilitation program	serum 25(OH)D, motor functional status, cognitive status	correlation between 25(OH)D and cognitive impairment ↑25(OH)D associated with greater functional gain	[146]
vit. D supplementation and rehabilitation outcomes in patients having hemiplegia—a randomized, double-blind, placebo-controlled study (*)	132 ischemic stroke patients hospitalized for 3-month hemiplegia rehabilitation	vit. D 300,000 IU or saline (i.m.), BRS, FAC, BBS-tests at the beginning and at the end of the rehabilitation program.	Significantly vit. D improved the BBS parameter	[147]
vit. D supplementation on stroke rehabilitation efficacy	76 patients receiving inpatient stroke rehabilitation treatment	weekly vit. D (50,000 IU, orally) for 4–12 weeks, BRS and FAC scores before and after rehabilitation	higher changes in FAC and BRS scores in patients receiving vit. D.	[148]
vit. D supplementation on rehabilitation after acute stroke (*)	100 patients after acute stroke multicentre, randomized, double-blind study	vit. D3 (2000 IU/day) or a placebo Barthel index scores	no significant differences between the groups	[149]
meta-analysis of association between vit. D status and the risk of stroke	19 studies ischemic and hemorrhagic stroke	circulating vit. D/vit. D intake	low vit. D level associated with ischemic stroke	[150]
associations of 25(OH)D and risk of poor functional outcome in nondiabetic stroke	266 nondiabetic Chinese stroke patients	serum 25(OH)D	VDD associated with an increased risk of poor functional outcome	[151]
vit. D, hypertension and ischemic stroke—Observational and genetic study	11,6655 Danish individuals genotyped for genetic variants in *DHCR7* and *CYP2R1*	serum 25(OH)D blood pressure, hypertension and ischemic stroke	*DHCR7* and *CYP2R1* allele score associated with lower 25(OH)D and higher blood pressure and hypertension	[153]
association of 25(OH)D with incident stroke REGARDS study	610 participants with incident stroke and 937 stroke-free individuals	serum 25(OH)D	low 25(OH)D associated with higher risk of stroke irresprective of black or white race	[154]
associations of 25(OH)D level and SNP status with incident stroke	12,158 participants in ARIC study	serum 25(OH)D level; DBP SNPs: rs7041, rs4588	low 25(OH)D level associated with higher stroke risk; possible associacion with DBP SNPs	[155]
association between vit. D metabolites, cognitive function and brain atrophy in elderly individuals	390 community-dwelling elderly individuals with normal neurological status and without history of stroke and dementia	serum 25(OH)D3, 25(OH)D2 and 24,25(OH)2D3	worse memory in individuals with low 25(OH)D and 24,25(OH)2D3	[157]
association between vit. D and the development of PSD	89 patients with acute ischemic stroke and 100 healthy controls	serum 25(OH)D	lower 25(OH)D level in non-PSD and PSD patients vs. healthy controls lower 25(OH)D level in PSD vs. non-PSD	[161]
association between vit. D and PSA	226 first acute ischemic stroke patients and 100 healthy subjects	serum 25(OH)D	low serum 25(OH)D level in PSA and non-PSA patients vs. healthy subjects significant association between PSA and low 25(OH)D	[162]
association between vit. D supplementation and rehabilitation after stroke—randomized double blind, parallel, monocentric clinical trial (*)	40 patients undergoing intensive neuro-rehabilitation treatment after stroke	cholecalciferol 2000 IU, oral/day/12 weeks,	no significant effect of vit. D supplementation on the beneficial effects of rehabilitation	[16]
association between 25(OH)D serum level with initial stroke severity and infarct volume	235 patients who were admitted within 24 h of acute ischemic stroke onset	serum 25(OH)D the volume of cerebral infarction	low 25(OH)D level in acute ischemic stroke as early predictor of larger infarct volume and neurological deficits	[144]
associations of vit. D with stroke recurrence in a 3-month follow-up study	349 Chinese patients with first-ever ischemic stroke	serum 25(OH)D	low serum 25(OH)D in patients with recurrent stroke	[140]
association between vit. D and rehabilitation in stroke patients	100 stroke patients	serum 25(OH)D	rehabilitation efficacy positively correlated with 25(OH)D level and negatively	[163]

↑: increase; ↓: decrease; AIS: acute ischemic stroke; BBS: Berg balance scale (to evaluate the balance and fall risk); BRS: Brunnstrom recovery stage (lower extremity motor function); *CYP2R1*: vitamin D 25-hydroxylase, *DHCR7*: 7-dehydrocholesterol reductase; FAC: functional ambulation classification; HIE: hypoxic-ischemic encephalopathy; hsCRP: high-sensitivity C-reactive protein; PSA: post-stroke anxiety; PSD: post-stroke depression; S100-B: Ca^2+^-binding cytosolic protein expressed in glial cells; VDD: vitamin D deficiency; *—randomized control studies.

## 7. Conclusions

Vitamin D3 is a multitarget drug with both genomic and non-genomic actions and well-evidenced neuroprotective, anti-inflammatory and regenerative properties. As shown by experimental studies, vitamin D3 can interfere with pivotal molecular mechanisms of neuronal death and survival in each phase of stroke. Moreover, besides its beneficial effect on neuronal tissue, it has a protective effect on the vascular system and BBB integrity (Figure 1). Therefore, it appears to be the right candidate for prevention of strokes, which are, in essence, cerebrovascular disorders, and for faster restoration of neurological functions in post-stroke patients during their rehabilitation. Apart from its beneficial effects on neurological conditions, vitamin D3 may diminish post-stroke depression and anxiety due to its stimulating effects on neurotransmitters and BDNF synthesis. Nevertheless, preclinical models of stroke are criticized because they do not take into account the most important risk factors for stroke, i.e., advanced age and co-morbidities, e.g., hypertension and diabetes [125]. The reasons for the mixed results of clinical trials of vitamin D3 supplementation in stroke patients could be due to problems with choosing the right doses of this secosteroid, unknown functional state of VDR in the treated patients, shorter or longer follow-up times, and various cutoff values for vitamin D deficiency. Polymorphisms of VDR and vitamin D3 synthesizing/metabolizing enzymes, as well as ethnic and sex-related factors might also interfere with obtaining conclusive results. Moreover, the molecular complexity of vitamin D3 mechanism of action, as well as its synergistic effects with progesterone and estrogens in protecting cells against ischemia-related injury suggest that also in clinical trials hormonal status of patients with stroke may be an important factor for vitamin D3 therapeutic effectiveness. Nevertheless, an opinion prevails that severe VDD might worsen outcomes of stroke. A better understanding of molecular mechanism of vitamin D3 and further developments in pharmacogenomics should help establish more precisely the effective doses and the right group (subpopulation) of stroke patients which could benefit from vitamin D3 treatment. It is not excluded that, in the future, the therapy will also be improved by development of new vitamin D analogues with increased neuroprotective activity and/or by targeted delivery of vitamin D3 or its mimics to the brain.

## Figures and Tables

**Figure 1 antioxidants-11-02120-f001:**
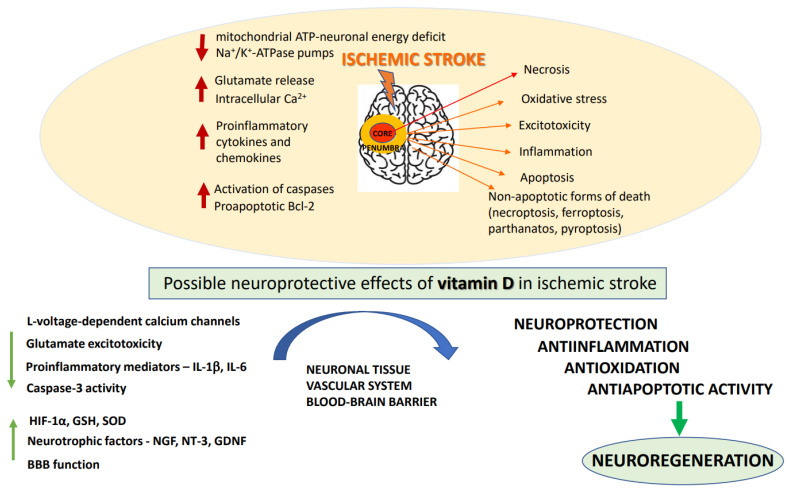
The pathogenic mechanisms of ischemic stroke and neuroprotective effects of vitamin D. BBB: blood–brain–barrier; GDNF: glial-derived neurotrophic factor; GSH: glutathione; HIF: 1a-hypoxia-inducible factor; IL: interleukins; NGF: nerve growth factor; NT: 3-neutrophin-3; SOD: superoxide dismutase.

**Table 1 antioxidants-11-02120-t001:** Effects of vitamin D in experimental stroke models—in vitro studies.

Model	Cell Treatment	Effects of Vitamin D	Reference
Glu and NMDA-induced excitotoxic damage in rat primary hippocampal neurons	1,25(OH)2D3 (0.1–1000 nM) NMDA (100 μM), Glu (5 μM)	1,25(OH)2D3 (1–100 nM) ↑cell viability ↓VGCC	[95]
Glu and dopaminergic toxins-induced rat mesencephalic cells damage	1,25(OH)2D3 (1–100 nM) Glu (1 mM) H_2_O_2_ (30 μM) Calcium ionophore (A23187, 1 μM) MPP+ (30 μM) 6-OHDA (100 μM)	↑cell viability ↓ROS	[96]
Glu-induced damage in primary rat cortical neurons	1,25(OH)2D3 (10–100 nM)/3–9 DIV Glu (100 μM)	↑cell viability ↑VDR mRNA ↑MAP-2 ↑GAP-43 ↑synapsin-1	[97]
Primary neocortical, hippocampal and cerebellar cell cultures exposed to Glu	1,25(OH)2D3 (50 and 100 nM)/30 min, 1, 3, 6 or 9 h after Glu (1 mM)	↓excitotoxicity ↓caspase-3 activity	[98]
Glu-induced primary cortical neurons damage	Progesterone (0.1-80 μM) and 1,25(OH)2D3 (1–100 nM) individually or in different combinations Glu (0.5 μM)	↓neuronal loss ↑p-ERK1/2	[99]
OGD model in primary cortical neurons	Progesterone (0.1–80 μM) and 1,25(OH)2D3 (0.001–5 μM) individually or in different combinations during OGD and reoxygenation	↓neuronal loss	[100]
Hypoxia model in C57BL/6J mice primary neuronal cells	cholecalciferol, 0.01–1 µM/14DIV, 20 min before hypoxia, during hypoxia and immediately after reoxygenation	cholecalciferol-1 µM ↓cell viability, ↓the neuron-glial functional structure cholecalciferol-0.01-0.1 µM ↑cell viability ↑the functional structure and activity of neuron–glial networks	[101]

↑: increase; ↓: decrease; 1,25(OH)2D3: calcitriol; DIV: days in vitro; ERK1/2: extracellular signal-regulated kinase 1 and 2; GAP-43: growth-associated protein-43; Glu: glutamate; L-VGCC: voltage-dependent calcium channels; MAP-2: microtubule-associated protein-2; MPP+: 1-Methyl-4-phenylpyridinium; OGD: oxygen glucose deprivation; 6-OHDA: 6-hydroxydopamine; ROS: reactive oxygen species; VDR: vitamin D receptor.

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
