# Peer review of "Vitamin D3 and Ischemic Stroke: A Narrative Review"

_antioxidants, 2022, doi:10.3390/antiox11112120_

Round 1
Reviewer 1 Report
This is a well written overinclusive narrative review on vitamin D and ischemic stroke or more broadly brain ischemia.
Only a few and minor comments: i) In the introduction is worth mentioning the current treatment of ischemic strokes which is the reperfusion therapies; ii) lines 61-63 need a reference; iii) I would suggest in table 3 randomized control studies to be marked with a star (or else) in order to have these studies stand out, as most are association studies.
Author Response
- „In the introduction is worth mentioning the current treatment of ischemic strokes which is the reperfusion therapies.” Response: The following sentence has been added to the Introduction: The current treatment of acute ischemic strokes is based on reperfusion therapies including intravenous administration of thrombolytic agents and endovascular therapy.
- lines 61-63 need a reference; Response: The following reference has been added: „Cui, X.; Gooch, H.; Petty, A.; McGrath, J. J.; Eyles, D. Vitamin D and the Brain: Genomic and Non-Genomic Actions. Mol. Cell. Endocrinol. 2017, 453, 131–143. https://doi.org/10.1016/j.mce.2017.05.035.”
- I would suggest in table 3 randomized control studies to be marked with a star (or else) in order to have these studies stand out, as most are association studies. Response: The randomized control studies have been marked as requested.
Reviewer 2 Report
Minor changes in yellow, strongly removed in red. See attached the suggested revisions highlighted.

Author Response
- Minor changes in yellow, strongly removed in red. See attached the suggested revisions highlighted.
Response: Thank you. All suggested by the reviewer changes have been introduced in the revised manuscript.
Reviewer 3 Report
Title of the review: Vitamin D3 and ischemic stroke: A narrative review
It is well known that Vitamin D3 has an extensive physiological role in humans. The present review covered the various roles of Vitamin D3 in ischemic stroke. The paper has explained collectively related mechanisms with support of in vitro, in vivo, and clinical literature. The review covers excellent scientific information and is presented in an effective manner. However, I have some minor suggestions to improve the presentation of the paper.
1. Referring to line numbers 45 and 46, the information on prevalence data is linked with early publications. Regarding prevalence, it would be good to follow the updated one.
2. In Page 3, Regarding the biochemical basis of ischemic stroke, it will be more interesting to the reader to add a diagrammatic illustration that contains the major pathogenic mechanisms like the effect of NMDA-related excitotoxicity due to higher levels of intracellular Ca2+, neuronal inflammation, oxidative stress, and cellular apoptosis.
3. Referring to line numbers 167-168, regarding proapoptotic Bcl-2 family members, the examples of apoptosis proteins are missing.
4. Referring to line numbers 179-180, need to be added examples of responsive anti-inflammatory cytokines.
5. Referring to line numbers 373-374, “Of note, the distribution of nVDR receptors in the human and rodent brain is very similar”, which needs to be a double confirmation.
6. Page number 22, refer to the table, “982 stroke patients (629 males, 357 females)”, need a confirmation about the number of patients.
7. There are a lot of errors in using symbols and extra space between the words, which may be related to some technical issues. However, need careful revision.
Author Response
- Referring to line numbers 45 and 46, the information on prevalence data is linked with early publications. Regarding prevalence, it would be good to follow the updated one.
Response: The information on prevalence data has been updated by adding the reference: Swanepoel, A. C.; Pretorius, E. Prevention and Follow-up in Thromboembolic Ischemic Stroke: Do We Need to Think out of the Box? Thromb. Res. 2015, 136 (6), 1067–1073. https://doi.org/10.1016/j.thromres.2015.11.001.
- In Page 3, Regarding the biochemical basis of ischemic stroke, it will be more interesting to the reader to add a diagrammatic illustration that contains the major pathogenic mechanisms like the effect of NMDA-related excitotoxicity due to higher levels of intracellular Ca2+, neuronal inflammation, oxidative stress, and cellular apoptosis.
Response: The diagrammatic illustration (Fig. 1) that contains the major pathogenic mechanisms of ischemic stroke has been added to the revised manuscript.
- Referring to line numbers 167-168, regarding proapoptotic Bcl-2 family members, the examples of apoptosis proteins are missing.
Response: The manuscript has been supplemented with examples of Bax, Bad and Bak proteins.
- Referring to line numbers 179-180, need to be added examples of responsive anti-inflammatory cytokines.
Response: Examples of responsive anti-inflammatory IL-4 and IL-10 have been added.
- Referring to line numbers 373-374, “Of note, the distribution of nVDR receptors in the human and rodent brain is very similar”, which needs to be a double confirmation.
Response: The similarity of the distribution of nVDR receptors in the human and rodent brain has been confirmed by immunohistochemical, western blotting and RT-PCR studies. The reference Cui, X.; Gooch, H.; Groves, N. J.; Sah, P.; Burne, T. H.; Eyles, D. W.; McGrath, J. J. Vitamin D and the Brain: Key Questions for Future Research. J. Steroid Biochem. Mol. Biol. 2015, 148, 305-309. https://doi.org/10.1016/j.jsbmb.2014.11.004 has been added.
- Page number 22, refer to the table, “982 stroke patients (629 males, 357 females)”, need a confirmation about the number of patients.
Response: The number of patients has been corrected. The number 982 has been replaced by 986.
- There are a lot of errors in using symbols and extra space between the words, which may be related to some technical issues. However, need careful revision.
Response: The manuscript has been carefully revised and the errors have been removed.
We thank you for your insightful comments.